# Protective Effects of Licochalcone A Ameliorates Obesity and Non-Alcoholic Fatty Liver Disease Via Promotion of the Sirt-1/AMPK Pathway in Mice Fed a High-Fat Diet

**DOI:** 10.3390/cells8050447

**Published:** 2019-05-11

**Authors:** Chian-Jiun Liou, Yau-Ker Lee, Nai-Chun Ting, Ya-Ling Chen, Szu-Chuan Shen, Shu-Ju Wu, Wen-Chung Huang

**Affiliations:** 1Department of Nursing, Division of Basic Medical Sciences, Research Center for Chinese Herbal Medicine, Chang Gung University of Science and Technology, No.261, Wenhua 1st Rd., Guishan Dist., Taoyuan City 33303, Taiwan; ccliu@mail.cgust.edu.tw; 2Division of Allergy, Asthma, and Rheumatology, Department of Pediatrics, Chang Gung Memorial Hospital, Linkou, Guishan Dist., Taoyuan City 33303, Taiwan; 3Graduate Institute of Health Industry Technology, Research Center for Food and Cosmetic Safety, College of Human Ecology, Chang Gung University of Science and Technology, No.261, Wenhua 1st Rd., Guishan Dist., Taoyuan City 33303, Taiwan; fsj880215@hotmail.com; 4Graduate Institute of Clinical Medical Sciences, Chang Gung University, No.259, Wenhua 1st Rd., Guishan Dist., Taoyuan City 33303, Taiwan; graceting1208@gmail.com; 5Department of Nutrition and Health Sciences, Research Center for Chinese Herbal Medicine, College of Human Ecology, Chang Gung University of Science and Technology, No.261, Wenhua 1st Rd., Guishan Dist., Taoyuan City 33303, Taiwan; ylchen01@mail.cgust.edu.tw; 6School of Nutrition and Health Sciences, Taipei Medical University, 250 Wu-Hsing Street, Taipei City 11031, Taiwan; 7Graduate Program of Nutrition Science, National Taiwan Normal University, 88 Ting-Chow Rd, Sec 4, Taipei City 11677, Taiwan; scs@ntnu.edu.tw; 8Aesthetic Medical Center, Department of Dermatology, Chang Gung Memorial Hospital, Linkou, Guishan Dist., Taoyuan 33303, Taiwan

**Keywords:** AMPK, HepG2, licochalcone A, lipolysis, obesity, nonalcoholic fatty liver disease

## Abstract

Licochalcone A is a chalcone isolated from *Glycyrrhiza uralensis*. It showed anti-tumor and anti-inflammatory properties in mice with acute lung injuries and regulated lipid metabolism through the activation of AMP-activated protein kinase (AMPK) in hepatocytes. However, the effects of licochalcone A on reducing weight gain and improving nonalcoholic fatty liver disease (NAFLD) are unclear. Thus, the present study investigated whether licochalcone A ameliorated weight loss and lipid metabolism in the liver of high-fat diet (HFD)-induced obese mice. Male C57BL/6 mice were fed an HFD to induce obesity and NAFLD, and then were injected intraperitoneally with licochalcone A. In another experiment, a fatty liver cell model was established by incubating HepG2 hepatocytes with oleic acid and treating the cells with licochalcone A to evaluate lipid metabolism. Our results demonstrated that HFD-induced obese mice treated with licochalcone A had decreased body weight as well as inguinal and epididymal adipose tissue weights compared with HFD-treated mice. Licochalcone A also ameliorated hepatocyte steatosis and decreased liver tissue weight and lipid droplet accumulation in liver tissue. We also found that licochalcone A significantly regulated serum triglycerides, low-density lipoprotein, and free fatty acids, and decreased the fasting blood glucose value. Furthermore, in vivo and in vitro, licochalcone A significantly decreased expression of the transcription factor of lipogenesis and fatty acid synthase. Licochalcone A activated the sirt-1/AMPK pathway to reduce fatty acid chain synthesis and increased lipolysis and β-oxidation in hepatocytes. Licochalcone A can potentially ameliorate obesity and NAFLD in mice via activation of the sirt1/AMPK pathway.

## 1. Introduction

Obesity is becoming prevalent in developed and developing countries and is considered an important risk factor for chronic diseases including cardiovascular diseases, hypertension, osteoarthritis, some cancers, and diabetes [1]. Fatty liver disease is mainly divided into alcoholic fatty liver disease and nonalcoholic fatty liver disease (NAFLD). In recent years, some studies found that obesity was an important factor causing NAFLD and hepatic steatosis [2]. Most patients with NAFLD also have metabolic diseases such as obesity, diabetes, and hyperlipidemia, and NAFLD can be divided into the nonalcoholic fatty liver and nonalcoholic steatohepatitis (NASH) stages according to the histology and disease development [3]. In the NASH stage, hepatic cells cause persistent inflammation, which leads to steatosis of the liver, chronic inflammation, and damage of hepatocytes [4]. If the patient does not adjust their diet and exercise properly, chronic inflammation of the liver would turn into liver cirrhosis or failure, and may even cause liver cancer.

Many studies confirmed that the CCAAT/enhancer-binding protein (C/EBP) and peroxisome proliferator-activated receptor (PPAR) were key transcriptional factors of lipogenesis in the liver and adipose tissue [5]. In the early stage of differentiation, adipocytes express PPAR-β and C/EBPα to assist with differentiation and then express PPAR-γ and C/EBPβ for enhanced lipid accumulation in the late stage of differentiation [6]. Other studies found that sterol regulatory element binding protein 1c (SREBP-1c) also regulated the expression of fatty acid synthase (FAS) to increase lipid synthesis in the liver or adipose tissue [5]. Thus, inhibiting the activity of transcription factors for lipid synthesis would improve the body weight of obese mice and reduce lipid accumulation in the liver and adipose tissue.

In recent years, the research identified that the sirtuin1(Sirt1)/AMP-activated protein kinase (AMPK) pathway was an important regulator of energy, which modulated lipid and glucose metabolism in hepatocytes [7]. Resveratrol is a sirt1 activator, which reduced body weight and ameliorated fatty liver by promoting sirt1 and AMPK activity to suppress lipogenesis [8]. Sirt1-deficient mice had inhibited AMPK activity and increased SREBP-1c expression that triggered hepatic steatosis and obesity [9]. Metformin, an antidiabetic drug, improved hepatic steatosis by increasing sirt1/AMPK activation and fatty acid β-oxidation in obese mice [10]. Activation of AMPK inhibited transcription factors of lipogenesis and FAS and promoted adipocyte differentiation [11]. Therefore, promoting the sirt1/AMPK pathway may inhibit lipid accumulation and NAFLD in obese mice.

Many studies found that green tea extract and *Psoralea corylifolia* extract could increase lipolysis in the liver of obese mice [12,13]. Those natural extracts could increase lipolysis of fatty liver cells and improve NAFLD by activating adipose triglyceride lipase (ATGL) and hormone-sensitive lipase (HSL) expression. Saponins, isolated from *Panax ginseng*, also prevent obesity via regulating lipolysis in obese mice [14]. Therefore, those natural compounds promote the decomposition of excessive lipid accumulation and can reduce body weight and improve fat accumulation in the liver.

Licochalcone A is a chalcone isolated from *Glycyrrhiza uralensis* Fisch, and has multiple anti-oxidant, anti-tumor, and anti-inflammatory activities [15,16,17]. Licochalcone A prevented adipocyte differentiation and lipogenesis via suppressed PPAR-γ and SREBP-1c expression in 3T3-L1 adipocytes [18]. Previous studies found that glycyrrhizic acid isolated from *G. uralensis* Fisch may be effective for treating NAFLD in mice [19]. Licochalcone A regulated lipid metabolism by activating AMPK in hepatocytes [20]. Licochalcone A could also protect LPS/d-galactosamine-induced acute liver injury via modulated Nrf2 and autophagy [21,22]. Licochalcone A was found to decrease the triglyceride levels in the liver of high-fat diet (HFD)-induced obese ICR mice [20]. However, it is unclear how licochalcone A improves NAFLD. In the present study, we evaluated whether licochalcone A could modulate lipid metabolism and molecular mechanisms in oleic acid-induced hepatocytes in vitro. We also examined whether licochalcone A could improve NAFLD, and suppress lipogenesis and lipolysis in HFD-induced obese mice.

## 2. Materials and Methods

### 2.1. Animals and Administration of Licochalcone A

Four-week-old male C57BL/6 mice were procured from the National Laboratory Animal Center in Taiwan. High-fat diet (No. D12492) was purchased from Research Diets, Inc. (Middlesex County, NJ, USA). Mice were randomly divided into the following four experimental groups (each group n = 12): (1) normal mice group (N): mice were fed normal chow diet and administered DMSO by intraperitoneal injection; (2) HFD mice group (HFD): mice were fed an HFD (contained 60% fat) for 16 weeks and administered DMSO by intraperitoneal injection; (3) LA5 and (4) LA10 mice groups: mice maintained an HFD and were administered 5 mg/kg and 10 mg/kg, respectively, licochalcone A (purity ≥98%, St. Louis, MO, USA) dissolved in DMSO by intraperitoneal injection. The HFD, LA5, and LA10 groups maintained an HFD for 4 weeks and were then treated twice a week for 12 weeks (Figure 1A). All animal experiments were approved by the Laboratory Animal Care Committee of Chang Gung University of Science and Technology (IACUC approval numbers 2017-002-V1).

### 2.2. Histological Analysis

Adipose and liver tissues were fixed with formalin and then embedded in paraffin. All tissues were sliced into 6-μm sections and stained with hematoxylin and eosin (HE) solution as previously described [23]. A periodic acid-Schiff (PAS) staining system (Sigma) was used to detect glycogen accumulation in liver tissue, as described previously [24]. All biopsy specimens were examined using a light microscope (Olympus, Tokyo, Japan). The NAFLD score was evaluated as previously described [25].

### 2.3. Immunohistochemistry (IHC)

Immunohistochemical staining of liver tissues for FAS, carnitine palmitoyltransferase I (CPT-1), and sirt1was performed using paraffin-embedded sections (6 μm). Each slide was incubated with the primary antibody (1:50) overnight, washed, and incubated with horseradish peroxidase anti-rabbit secondary antibody. The slides were treated with DAB substrate to detect specific protein expression using a light microscope (Olympus).

### 2.4. Biochemical Analysis

Serum was collected and a biochemistry analyzer (DRI-CHEM NX500, Fuji, Tokyo, Japan) was used to assay the levels of total triglycerides (TG), total cholesterol (TC), total bilirubin, high-density lipoprotein (HDL), low-density lipoprotein (LDL), glutamate oxaloacetate transaminase (GOT), and glutamate pyruvate transaminase (GPT) according to the manufacturer’s instructions. Free fatty acid was measured using a fatty acid quantitation kit (Sigma) according to the manufacturer’s protocol. Furthermore, the day before the end of the animal experiment, mice were fasted for 16 h and received an intraperitoneal injection of glucose to detect the levels of glucose using a biochemistry analyzer (Fuji) and levels of insulin using the insulin EIA Kit (Cayman, Ann Arbor, MI, USA). The serum TNF-α was detected using specific ELISA kits (R&D, Minneapolis, MN, USA).

### 2.5. Western Blot Analysis

Protein extracts were prepared using a protein lysis kit (Sigma) and then separated on 8–10% SDS–PAGE gels. Next, gels were transferred to polyvinylidene difluoride (PVDF) membranes and incubated with primary specific antibodies overnight. The PVDF membrane was washed and incubated with secondary antibodies at room temperature for 1 h. Finally, Luminol/Enhancer solution (Millipore, Billerica, MA, USA) was added to detect specific protein expression using the BioSpectrum 600 system (UVP, Upland, CA, USA). Primary specific antibodies included β-actin (Sigma), ATGL, HSL, phosphorylated HSL (pHSL), C/EBPα, C/EBPβ, PPAR-α, PPAR-γ, acetyl CoA carboxylase-1 (ACC-1), phosphorylated-ACC-1 (pACC-1), (Abcam, Cambridge, MA, USA), CPT-1, CPT2, AMPKα, phosphorylated AMPKα (pAMPKα), SREBP-1c, FAS, and sirt1, (Cell Signaling Technology, Danvers, MA, USA).

### 2.6. RNA Isolation and Real-Time PCR for Gene Expression

Liver tissues were homogenized and total RNA was extracted for cDNA synthesis using a cDNA synthesis kit (Life Technologies), as previously described [26]. Specific gene expression levels were detected by labeled fluorescent SYBR Green and amplified using a spectrofluorometric thermal cycler (iCycler; Bio-Rad Laboratories, Hercules, CA, USA). The specific primers were shown in Table 1.

### 2.7. Cell Culture and Induced Fatty Liver Cells

The HepG2 hepatocyte cell line was purchased from the Bioresource Collection and Research Center (BCRC, Taiwan), and cultured at 37 °C in a 5% CO_2_ atmosphere in DMEM medium (Invitrogen-Gibco^TM^, Paisley, Scotland) supplemented with penicillin/streptomycin and 10% FBS. Oleic acid solution was purchased from Sigma, Inc (St. Louis, MO, USA). HepG2 cells were incubated with 0.5 mM oleic acid to stimulate lipid accumulation for 48 h; then cells were treated with licochalcone A (0–12 μM) for 24 h to evaluate the mechanism of lipid metabolism.

### 2.8. Cell Viability Assay

Licochalcone A was dissolved in DMSO. For all cell experiments, less than 0.1%. DMSO was used. HepG2 cells were incubated with various concentrations of licochalcone A for 24 h to detect cell viability using MTT solution (Sigma). Next, the culture plates were treated with isopropanol to evaluate cell viability using a spectrophotometer (Multiskan FC).

### 2.9. Oil Red O Staining

HepG2 cells were seeded and grown in 6-well plates and stimulated with 0.5 mM oleic acid for 48 h. Then, those cells were treated with various concentrations of licochalcone A (0–12 μM) for 24 h. Cells were harvested and fixed with formalin and stained with oil red O solution (Sigma) to detect oil droplets, as described previously [27]. Oil droplets were observed and images were obtained using an inverted microscope (Olympus).

### 2.10. The Effect of Licochalcone A on Hepatic Fatty Acid Uptake

HepG2 hepatocytes were stimulated with 0.5 mM oleic acid for 48 h and then cells were treated with licochalcone A for 24 h before staining with the fluorescent probe BODIPY FL C12 (Invitrogen, Carlsbad, CA, USA) to detect fatty acid uptake by fluorescence microscopy (Olympus).

### 2.11. The Effect of Licochalcone A on Hepatic Lipid Accumulation and Lipoperoxidation

HepG2 hepatocytes were stimulated with 0.5 mM oleic acid for 48 h and then cells were treated with licochalcone A for 24 h before staining with the fluorescent probes BODIPY 581/591 C11 and BODIPY 493/503 (Invitrogen) to detect lipoperoxidation and lipid accumulation, respectively, as described previously [28]. Moreover, cell nuclei were stained with DAPI and all fluorescent images were observed with a fluorescence microscope (Olympus).

### 2.12. Statistical Analysis

Statistical analyses were performed using a one-way analysis of variance (ANOVA) and a Dunnett post hoc test. All data are expressed as the mean ± SEM. *p*-values < 0.05 were considered significant.

## 3. Results

### 3.1. Licochalcone A Reduced Body Weight in Obese Mice

Mice were fed an HFD to induce obesity and administered DMSO or licochalcone A by intraperitoneal injection for 12 weeks. Mice were weighed twice a week, and we found that the weight of HFD mice gradually increased compared with normal mice. Surprisingly, mice treated with licochalcone A weighed significantly less than HFD mice during the experiment stage (Figure 1B). In the last weeks of the experiment, visual observation showed that LA5 and LA10 mice had significantly attenuated body weights compared with HFD mice (Figure 1C), and in the last week of the experiment, LA5 and LA10 mice had gained significantly less weight than HFD mice (LA5, 17.24 ± 2.35 g, *p* < 0.05; LA10, 13.15 ± 2.41 g, *p* < 0.01 vs. HFD, 21.35 ± 3.45 g; Figure 1D). Furthermore, LA5 and LA10 groups did not alter food intake compared to HFD mice (Figure 1E).

### 3.2. The Effect of Licochalcone A on the Weight of Adipose Tissue

Mice treated with licochalcone A had significantly reduced epididymal and inguinal adipose tissue weights compared with HFD mice (Figure 2A,B,E,F). Histological staining demonstrated that mice treated with licochalcone A had significantly decreased adipocyte sizes compared with HFD mice (Figure 2C,D,G,H).

### 3.3. The Effect of Licochalcone A on Liver Steatosis in Obese Mice

Grossly, the liver tissue was dark brown/red in normal mice; while, it was yellowish and lacked luster in HFD mice. We found that liver tissues could recover, as indicated by the dark brown/red color in LA5 and LA10 mice (Figure 3A). Furthermore, the liver weight of HFD mice was significantly increased compared with the liver weight of normal mice. Interestingly, licochalcone A could eliminate the liver weight gain seen in HFD obese mice (Figure 3B). Using HE staining of liver histological slices, we found lipid droplets and fat vacuoles were increased in HFD mice compared with normal mice and licochalcone A could significantly lessen the fat vacuoles and lipid accumulation in the liver of obese mice (Figure 3C,D). Glycogen was detected in hepatocytes by PAS staining. The hepatocytes of the HFD mice had significantly decreased glycogen accumulation and mice treated with licochalcone A had increased glycogen accumulation compared with HFD mice (Figure 3E). Licochalcone A also reduced TC and TG values of the liver in HFD mice (Figure 3F,G). We calculated lipid accumulation, ballooning, and lobular inflammation of liver tissue to evaluate NAFLD scores, and found that licochalcone A significantly ameliorated the NAFLD score compared with that of HFD mice (Figure 3F). Furthermore, we also assayed GOT and GPT levels in serum and found that licochalcone A could decrease GOP and GPT levels in treated mice compared with the HFD group mice (Table 2).

### 3.4. Effects of Licochalcone A on Serum Lipid Metabolism and Glucose Levels

Serum analysis showed that licochalcone A could significantly reduce TC, TG, free fatty acid, and LDL levels in HFD mice (Table 2). However, licochalcone A increased HDL and bilirubin levels compared with the HFD group. Furthermore, licochalcone A significantly attenuated the levels of leptin and increased adiponectin expression compared with HFD mice (Table 2). We also estimated insulin sensitivity to evaluate fasting glucose and insulin and found that the administration of licochalcone A could significantly reduce the levels of glucose and insulin compared with HFD mice. Licochalcone A significantly also inhibited HOMA-IR and TNF-α levels in obese mice (Table 2).

### 3.5. Licochalcone A Regulated Adipogenesis in Liver Tissue

Specific liver proteins were detected and we found that licochalcone A could suppress transcription factor SREBP-1c, PPAR-γ, and FAS expression compared with the HFD group (Figure 4A,E). Licochalcone A could also enhance phosphorylation of HSL and ATGL expression compared with the HFD group (Figure 4B,E). Moreover, when evaluating the fatty acid β-oxidation pathway of liver tissue, we found that administration of licochalcone A promoted CPT-1, but not CPT2, compared with the HFD group (Figure 4C,F), and licochalcone A also stimulated sirt1, phosphorylation of AMPKα, and phosphorylation of ACC-1 expression compared with the HFD group (Figure 4D,F). We also evaluated the expression of genes involved in lipogenesis, and found that licochalcone A could alleviate C/EBPα, Srebp-1c, FAS, and leptin, and increase CPT-1, HSL, sirt1, and adiponectin expression compared with HFD mice (Figure 5). However, licochalcone A did not affect CPT-2 gene expression compared with HFD mice. In liver tissue, IHC staining demonstrated licochalcone A treatment resulted in significantly recovered CPT-1 and sirt1 expression and decreased FAS production compared with the HFD group (Figure 6).

### 3.6. Cell Viability of HepG2 Cells Treated with Licochalcone A

We also investigated whether licochalcone A regulated lipid metabolism of hepatocytes in vitro. An MTT assay determined the cytotoxicity of licochalcone A in HepG2 cells. There was no cell cytotoxicity at licochalcone A concentrations ≤12.5 μM (data not shown); therefore, 1.5–12 μM licochalcone A was evaluated in all cell experiments.

### 3.7. Effects of Licochalcone A on Lipid Accumulation and Lipoperoxidation in HepG2 Cells

Staining with the fluorescent dye BODIPY 493/503 demonstrated that incubating HepG2 cells with oleic acid-induced lipid accumulation and licochalcone A significantly inhibited the accumulation of lipid droplets (Figure 7A,B). We also used oil red O stain to confirm that licochalcone A alleviated lipid droplets compared with oleic acid-induced HepG2 cells (Figure 7C). Hepatic lipoperoxidation was detected by BODIPY 581/591 C11, and we found that licochalcone A suppressed lipoperoxidation compared with oleic acid-induced HepG2 cells (Figure 8A,B). The BODIPY FL C12 fluorescent probe detected fatty acid uptake and results demonstrated that 12 uM licochalcone A could reduce block fatty acid uptake compared with the oleic acid-induced hepatocytes (Figure 8C).

### 3.8. Effect of Licochalcone A on Lipid Metabolism in Hepatocytes

Licochalcone A decreased SREBP-1c and FAS expression, involved in lipid synthesis, in fatty liver cells in vitro (Figure 9A). Licochalcone A also promoted ATGL, phosphorylation of HSL (Figure 9B), and CPT-1 expression compared with oleic acid-induced hepatocytes (Figure 9C). In addition, licochalcone A increased sirt1, phosphorylation of AMPK, and phosphorylation of ACC in fatty liver cells (Figure 9D). Furthermore, licochalcone A could recover the levels of phosphorylated AMPK, Sirt1, phosphorylated ACC, and ATGL and decrease FAS expression when oleic acid-induced HepG2 cells were co-treated with compound C (an AMPK inhibitor) (Figure 10A,B). Interestingly, oleic acid-induced HepG2 co-cultured with licochalcone A and AICAR (an AMPK activator) also had increased levels of phosphorylated AMPK, Sirt1, phosphorylated ACC, and ATGL, and decreased FAS expression compared with oleic acid-induced HepG2 cells treated with AICAR (Figure 10C,D).

## 4. Discussion

In developed and developing countries, the prevalence of obesity is increasing every year. Obesity induces high blood pressure, arteriosclerosis, high blood lipids, and also causes NAFLD, diabetes, cerebrovascular diseases, gout, degenerative arthritis, and cancer [29]. Therefore, obesity is considered a higher level risk factor during the development of chronic diseases. Normal liver cells store a lot of glycogen and small amounts of lipids to maintain physiological function [30]. However, when there is a fatty liver or NAFLD, the liver accumulates large amounts of lipids, which interfere with the physiological functions of the liver to block lipid and carbohydrate metabolism [2]. Therefore, patients have an abnormal metabolic microenvironment, and this causes insulin resistance and metabolic syndrome symptoms. In recent years, the goal of ameliorating and treating NAFLD is mainly to reduce lipid synthesis and lipid accumulation, and enhance the decomposition of fatty acids in fatty liver cells for promoting normal metabolism [13,23].

In recent years, many studies found that some plant extract or pure compounds could reduce body weight and improve NAFLD [1,8,31,32]. Rutin could improve NAFLD by suppressing hepatic lipid levels and oxidative injury in mice [33], and curcumin could ameliorate the progression of NASH and liver damage and in HFD mice by regulating the HMGB1-NF-κB pathway [34]. In HFD-induced obese mice, celastrol and resveratrol ameliorated body weight and reduced hepatic metabolic damage by promoting Sirt1 expression [8,9]. Scopolin, isolated from *Santolina oblongifolia*, could attenuate NAFLD and improve hepatic steatosis by facilitating the sirt1 and LKB1/AMPK signaling pathway in the liver of obese mice [35]. In this current study, licochalcone A could effectively reduce body weight and epididymal and inguinal adipose weights. Licochalcone A also decreased the liver weight, decreased lipid accumulation in the liver, and ameliorated NAFLD in HFD-fed obese mice. We also confirmed that licochalcone A effectively suppressed lipogenesis, and increased lipolysis and fatty acid β-oxidation by promoting the sirt1/AMPK pathway of the liver in vivo and in vitro. The root of *G. uralensis* Fischer is used to treat fever, cough, and pain in Chinese medicine [15]. *G. uralensis* extract also improved alcohol-induced fatty liver disease [36]. Clinically, licochalcone A maybe has potential as a novel, anti-obesity agent for treating NAFLD.

Obese people hoard visceral and excessive subcutaneous fat, move with difficulty, and have an increased incidence of chronic disease [37]. Adipose tissue in obese people secretes more inflammatory cytokines and chemokines to attract macrophages, which also activate and release more inflammatory cytokines to induce inflammation and insulin resistance in the adipose tissue and liver, leading to diabetes and metabolic syndrome [38]. Previous studies found that licochalcone A could block 3T3-L1 cell differentiation and reduce lipogenesis [18]. However, the effect of licochalcone A on reducing weight gain is unclear. In this current study, male C57BL/6 mice were fed an HFD to induce obesity, and licochalcone A effectively alleviated the body weight, adipose tissue weight, and adipocyte size in HFD obese mice. Therefore, licochalcone A has the potential to ameliorate obesity in HFD-induced obese mice. In addition, we found that licochalcone A can improve fasting blood glucose, and blood insulin values, ameliorating insulin resistance in obese mice. We plan to explore further whether licochalcone A has the potential to reduce the homeostatic model assessment for insulin resistance values and ameliorate diabetes in obese mice in future studies. The adipose tissue in obese mice can secrete excessive leptin to bind to the leptin receptor of hypothalamic neurons for reduced appetite and extenuated body weight [39]. Previous studies showed that resveratrol could improve obesity and insulin resistance via regulated leptin levels in serum [40]. Licochalcone A could significantly suppress the levels of leptin in serum and decrease leptin gene expression in the liver of obese mice. Hence, licochalcone A should be able to prevent diabetes in obese mice effectively.

Adiponectin is secreted by adipocytes and can regulate the levels of glucose and lipids in the serum [41]. Patients with hyperinsulinemia and type 2 diabetes have lower serum adiponectin concentrations than normal individuals [42]. Treatment with curcumin could enhance the levels of adiponectin to increase glucose metabolism, suppress lipogenesis, and improve insulin sensitivity in liver tissue [43]. Licochalcone A could significantly attenuate serum leptin levels and promote serum adiponectin levels to reduce obesity and regulate insulin sensitivity in HFD-induced obese mice. Moreover, we also measured the HOMA-IR value to assess insulin resistance. Licochalcone A significantly reduced HOMA-IR value for ameliorated insulin resistance in obese mice. Interestingly, licochalcone A also significantly decreased the weight of epididymal and inguinal adipose tissue and did not influence food intake compared with the intake of HFD mice.

Mice who are obese due to an HFD are not only suitable for observing the effects of obesity, but also for studying NAFLD [2]. The NAFLD score is based on macrophage infiltration, fat vacuoles, and blood biochemical indicators [25]. Previous studies showed that licochalcone A could reduce the triglyceride levels in HFD-induced obese mice [20]. However, how licochalcone A improves NAFLD and modulates the molecular mechanism of lipogenesis in obese mice is elusive. We found that HFD-induced obese mice had higher NAFLD scores, and treatment with licochalcone A reduced the NAFLD scores and decreased the levels of ALT and AST in serum to significantly ameliorate liver damage and inflammation indexes compared with obese mice. However, previous studies pointed out that an HFD did not significantly induce macrophage aggregation and related inflammation in the liver tissue of obese mice. Hence, we did not confirm whether licochalcone A could decrease macrophage infiltration in our experimental model. A methionine-choline-deficient diet could be used to induce a NASH mouse model and macrophage infiltration in the liver tissue [44]. In addition, a methionine-choline-deficient diet could be used to explore whether licochalcone A could suppress macrophage infiltration into liver tissue in future studies.

Activation of the AMPK pathway could regulate lipid and carbohydrate metabolism in liver cells [45]. Previous studies found that resveratrol could enhance sirt-1 and AMPK expression and reduce lipid accumulation in fatty liver cells [8]. In addition, AMPK-deficient mice have increased body weight and adipose tissue weight, and sirt1-deficient mice have increased body weight and NAFLD development [46]. Hence, the AMPK/sirt-1 pathway is closely related to NAFLD development. Licochalcone A increased protein expression of sirt1 and phosphorylated AMPK in the liver cells of HFD-induced obese mice. Furthermore, licochalcone A promoted sirt1 expression and phosphorylation of AMPK in oleic acid-induced HepG2 hepatocytes in vitro. The AMPK phosphorylation stimulates ACC phosphorylation, which blocks FAS expression for reduced fatty acid synthesis [45]. We confirmed licochalcone A increased ACC phosphorylation and inhibited FAS expression compared with the HFD group. In vitro cells co-cultured with licochalcone A and an AMPK inhibitor had recovered levels of phosphorylated ACC and decreased FAS. Hence, licochalcone A could block fatty acid synthesis in fatty liver cells. To understand the importance of the AMPK pathway, we also evaluated licochalcone A co-cultured with an AMPK enhancer (AICAR) in vitro in oleic acid-induced HepG2 cells. We found licochalcone A could increase sirt1expression, phosphorylation of AMPK, and ATGL production, and reduce FAS production. Hence, licochalcone A significantly improved lipogenesis and lipolysis of NAFLD by activating the AMPK pathway in HFD-induced obese mice. Furthermore, PAS staining demonstrated that licochalcone A was able to recover the glycogen accumulation in liver tissue that was decreased in HDF-induced obese mice. Hence, licochalcone A regulated glycogen synthesis and lipid accumulation and maintained the metabolic function in the liver.

The uptake of free fatty acids by liver cells would initiate transcriptional factors of lipogenesis to activate FAS for increased TG synthesis, leading to a fatty liver [5]. In this current study, mice treated with licochalcone A had significantly decreased SREBP-1c and PPAR-γ expression, leading to suppressed FAS expression and TG synthesis compared with HFD mice. Oleic acid-induced HepG2 cells treated with licochalcone A also had decreased SREBP-1c and FAS expression. Hence, licochalcone A could significantly decrease lipid droplet accumulation in oleic acid-induced HepG2 cells. Interestingly, fatty acid uptake, detected with the BODIPY FL C12 fluorescent probe, was inhibited with licochalcone treatment in oleic acid-induced hepatocytes. Previous studies have shown that free fatty acids inhibited AMPK activity and promoted transcriptional factors of lipogenesis in hepatocytes [47]. In this current study, cells co-cultured with licochalcone A and an AMPK inhibitor confirmed that licochalcone A could decrease FAS expression via the activated AMPK pathway. Thus, licochalcone A not only suppressed lipid synthesis in hepatocytes, but also blocked fatty acid uptake to decrease lipid accumulation through activation of the AMPK pathway.

Accelerating the lipid breakdown of hepatocytes also prevents the development of NAFLD [48]. Regular exercise and a balanced diet can promote lipid breakdown and modulated metabolic function in the liver [49]. Previous studies found that curcumin inhibited lipid synthesis and increased lipolysis in high-glucose-induced fatty liver cells and ameliorated lipolysis in the liver tissue of NAFLD mice [34]. Our study found that licochalcone A can improve the body weight of obese mice, reduce inguinal and epididymal adipose tissue weight, and adipocyte size. Licochalcone A also increased ATGL and phosphorylation of HSL, which promote lipolysis in the liver of obese mice. Hence, licochalcone A reduced fat vacuoles and the total weight of the liver via promotion of lipolysis in the NAFLD mice. In addition, licochalcone A treatment led to the liver returning to a dark red color compared with the yellowish livers of HFD-induced obese mice. AMPK activation could inhibit FOXO1 phosphorylation from promoting ATGL expression for lipolysis [50]. Our results with cells co-cultured with an AMPK inhibitor and licochalcone A confirmed that licochalcone A could enhance the lipolysis of fatty liver cells through activation of the AMPK pathway.

Free fatty acids cause an inflammatory response, and macrophages release more TNF-α to induce insulin resistance in adipose and liver tissues [51]. The metabolism of TGs in the liver produce free fatty acids that need to be rapidly decomposed by β-oxidation to produce energy or the formation of bile acids that are excreted through feces [48]. Many studies confirmed that AMPK activation and sirt1 expression enhanced CPT-1, which converts long-chain fatty acyl-CoA, promotes β-oxidation, and increases fatty acid decomposition [52]. Licochalcone A significantly increased CPT1 expression in the livers of HFD-induced mice. Resveratrol could enhance sirt1/AMPK expression to suppress ACC activity, increase β-oxidation for lipolysis, and improve NAFLD in HFD induced mice [8,40]. However, fatty acid decomposition causes oxidative damage in liver tissue. Previous studies found that silymarin could improve NAFLD via suppressed inflammatory and oxidative damage [53]. Licochalcone A was found to decrease oxidative stress injury by promoting the sirt1/Nrf2 pathway in rat primary cortical neurons [54]. We found that licochalcone A suppressed lipid peroxidation in oleic acid-induced fatty liver cells. Thus, licochalcone A has the potential to promote fatty acid decomposition and reduce oxidative stress injury in fatty liver cells.

Previous studies found that cholesterol transported into the liver would be converted to LDL and HDL [55,56]. Excessive accumulation of plasma LDL maybe forms plaques in the vascular wall for causing atherosclerosis. HDL can remove LDL and other lipoproteins from the circulation for decreasing the risk of cardiovascular disease [56]. Our results demonstrated that licochalcone A reduced TC, and LDL levels in serum of obese mice. Hence, licochalcone A can attenuate cardiovascular disease in obese mice. Bilirubin is the end product of heme metabolism. Some studies found that bilirubin could increase antioxidant capacity for improved oxidative stress-induced diseases [57]. In recent years, the research identified that the bilirubin could ameliorate insulin resistance by modulating cholesterol metabolism in obese mice [58]. Our experiment demonstrated that licochalcone A could restore serum bilirubin levels in obese mice. Licochalcone A also reduced the HOMA-IR value for improved insulin resistance. We thought that licochalcone A enhanced the bilirubin levels of the serum to protect the development of insulin resistance in obese mice.

## 5. Conclusions

In conclusion, we confirmed that licochalcone A reduced adipose tissue and body weights, and significantly reduced lipid accumulation in the liver of obese mice by promoting the sirt1/AMPK pathway, ameliorating hepatic steatosis. Therefore, licochalcone A has potential as a novel anti-obesity agent for the treatment of NAFLD.

## Figures and Tables

**Figure 1 cells-08-00447-f001:**
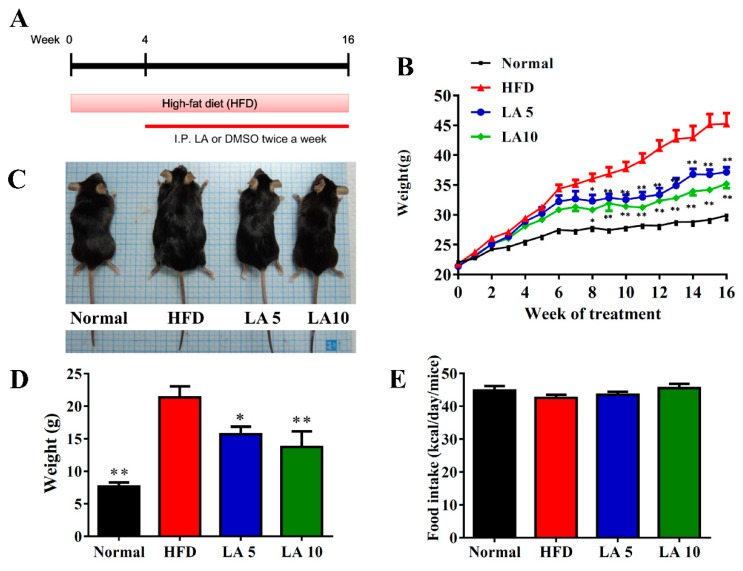
Licochalcone A (LA) reduced body weight in high-fat diet (HFD)-induced obese mice. (**A**) Male mice were fed an HFD (containing 60% fat) for 16 weeks, and administered DMSO, 5 mg/kg licochalcone A (LA5), or 10 mg/kg licochalcone A (LA10) by intraperitoneal injection (I.P.) twice a week from week 4 to 16. (**B**) Weight gain was measured for 16 weeks. (**C**) The appearance of the animal and (**D**) weight gain measured in the last week. (**E**) Food intake monitored each day. Data are presented as the mean ± SEM; *n* = 12. * *p* < 0.05, ** *p* < 0.01 compared with mice with HFD-induced obesity.

**Figure 2 cells-08-00447-f002:**
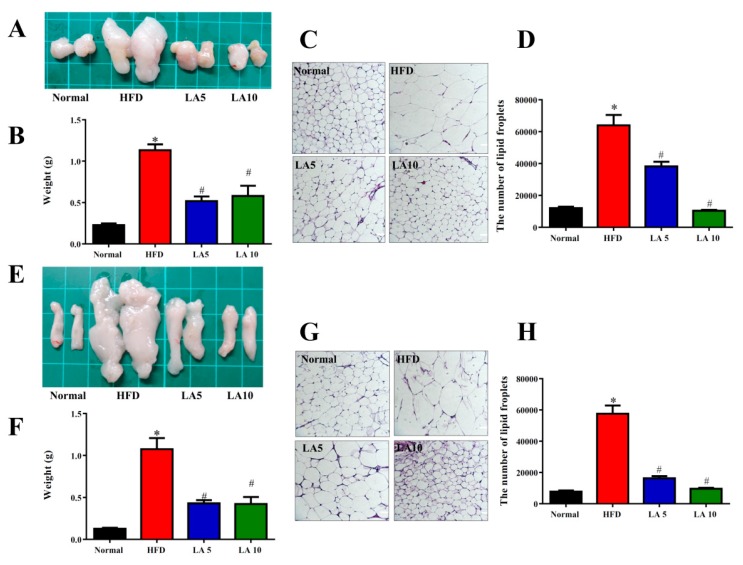
Licochalcone A (LA) reduced the epididymal and inguinal adipose tissue weights in HFD-induced obese mice. (**A**) The appearance and (**B**) weight of epididymal adipose tissue. (**C**) HE staining of epididymal adipose tissue (200× magnification). (**D**) The adipocyte size in epididymal adipose tissue. (**E**) The appearance and (**F**) weight of inguinal adipose tissue. (**G**) HE staining of inguinal adipose tissue (200× magnification). (**H**) The adipocyte size in inguinal adipose tissue. Data are presented as the mean ± SEM; *n* = 12. * *p* < 0.05, ** *p* < 0.01 compared with mice with HFD-induced obesity.

**Figure 3 cells-08-00447-f003:**
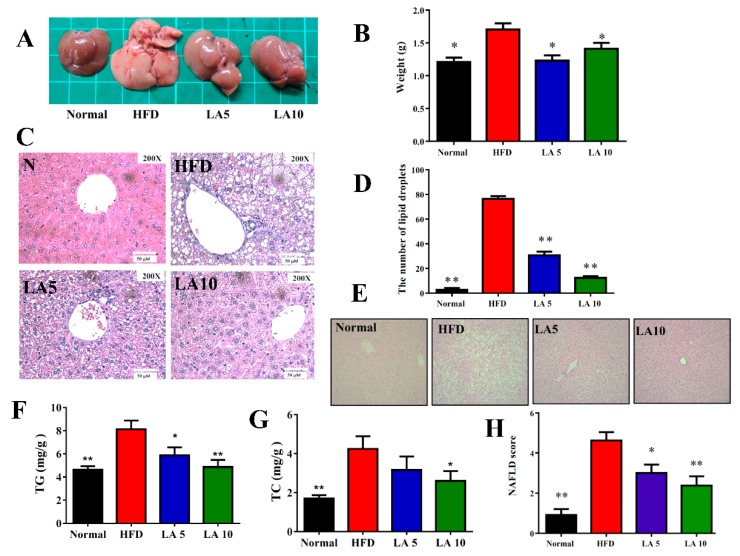
Licochalcone A (LA) ameliorated hepatic steatosis in HFD-induced obese mice. (**A**) The appearance of the liver, (**B**) liver weight, and (**C**) HE staining of liver tissues (200× magnification). (**D**) The calculated number of lipid droplets in liver tissue. (**E**) PAS staining demonstrating the glycogen distribution in the liver (200× magnification). (**F**) TG and (**G**) TC levels in the liver. (**H**) NAFLD scores based on the evaluation of HE staining of liver tissues. Data are presented as the mean ± SEM; *n* = 12. * *p* < 0.05, ** *p* < 0.01 compared with mice with HFD-induced obesity.

**Figure 4 cells-08-00447-f004:**
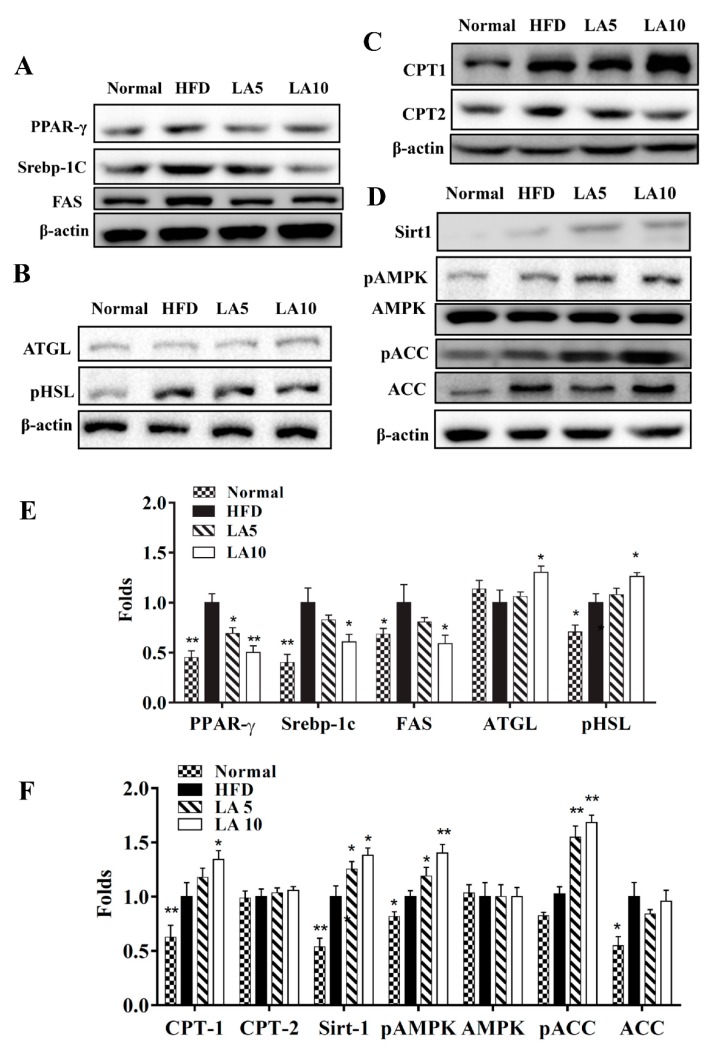
Effects of licochalcone A (LA) on lipid metabolism in mouse liver tissue. (**A**) Expression of transcription factors associated with adipogenesis and lipogenesis, (**B**) lipolysis, (**C**) β-oxidation, and (**D**) the sirt-1/AMPK pathway were detected by Western blot. (**E**,**F**) The fold expression levels were measured relative to the expression of β-actin. Three independent experiments were analyzed using β-actin as an internal control. The data are presented as the mean ± SEM. * *p* < 0.05, ** *p* < 0.01 compared to the HFD group.

**Figure 5 cells-08-00447-f005:**
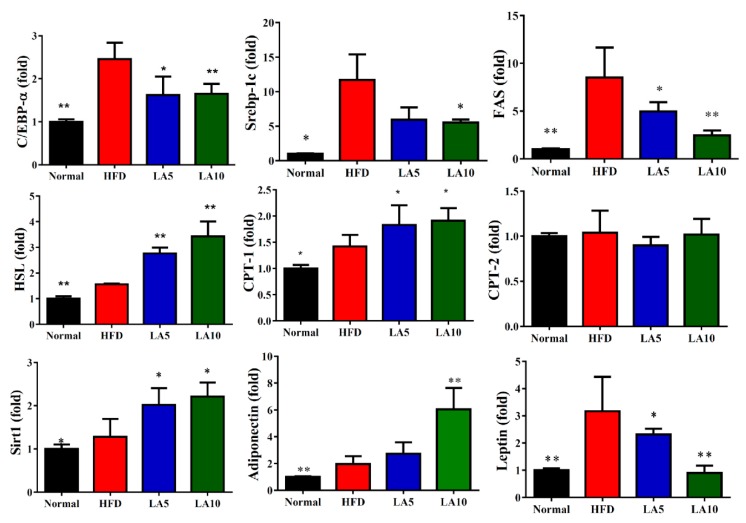
Licochalcone A (LA) modulated lipogenesis and lipolysis gene expression in liver tissue. Gene expression levels were determined using real-time PCR. * *p* < 0.05, ** *p* < 0.01 compared with mice with HFD-induced obesity.

**Figure 6 cells-08-00447-f006:**
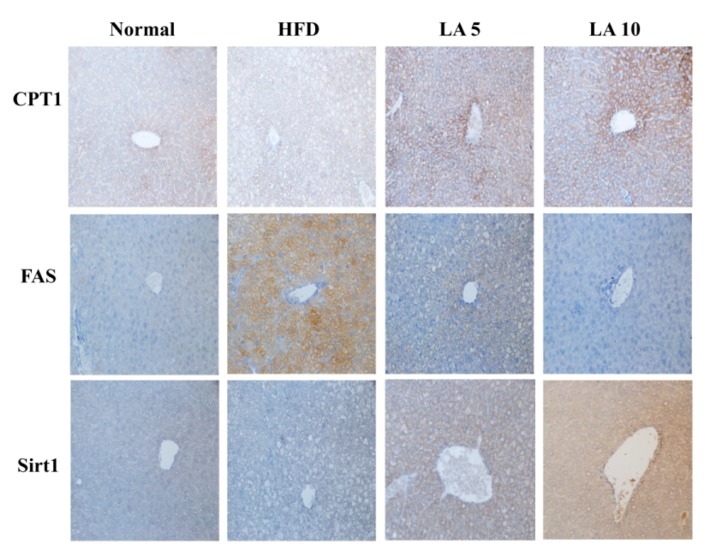
Licochalcone A (LA) modulated FAS, sirt-1, and CPT-1 expression in the liver. Expression levels of CPT-1, sirt1, and FAS were analyzed by immunohistochemistry and labeled as a brown color drop.

**Figure 7 cells-08-00447-f007:**
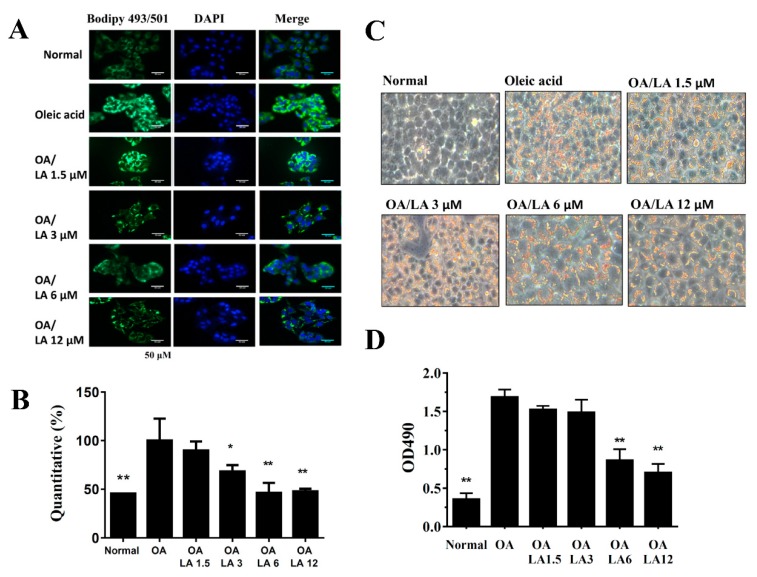
Licochalcone A (LA) reduced lipid accumulation in HepG2 cells. HepG2 cells were treated with 0.5 mM oleic acid (OA) at 37 °C for 48 h to induce lipid accumulation in hepatocytes, followed by treatment with licochalcone A (1.5–12 μM) for 24 h. (**A**) The fluorescent dye BODIPY 493/503 (green) was used to detect hepatic lipid droplets using a fluorescent microscope. Three independent experiments were analyzed. Nuclei were stained with DAPI (blue), and (**B**) Fluorescent images were quantified. (**C**) Oil red O staining showed lipid accumulation. (**D**) HepG2 cells treated with isopropanol and lipid accumulation measured using the absorbance at OD 490 nm. Three independent experiments were analyzed. The data are presented as the mean ± SEM; * *p* < 0.05, ** *p* < 0.01 compared with oleic acid-induced HepG2 cells.

**Figure 8 cells-08-00447-f008:**
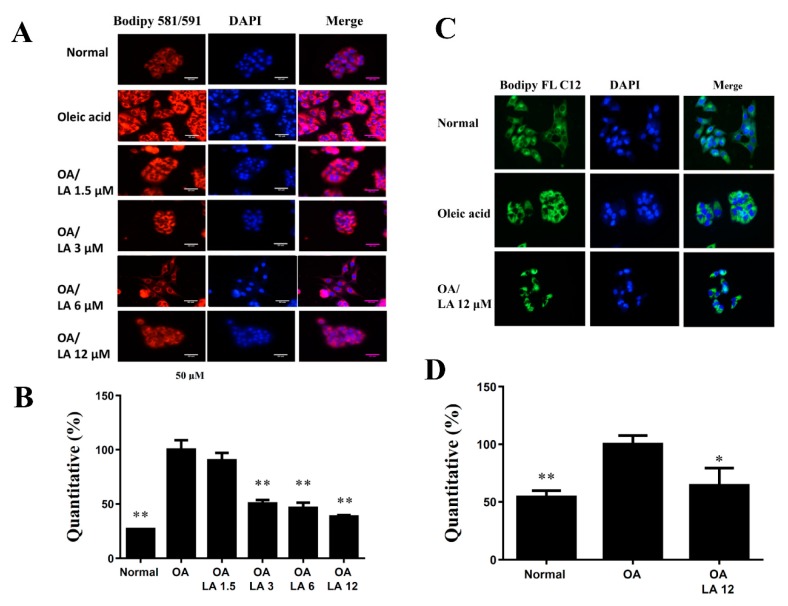
Licochalcone A (LA) reduced lipoperoxidation and fatty acid uptake into HepG2 cells. (**A**) HepG2 cells were treated with 0.5 mM oleic acid (OA) at 37 °C for 48 h to induce lipid accumulation in hepatocytes, followed by treatment with licochalcone A (1.5–12 μM) for 24 h. The fluorescent dye BODIPY 581/591 C11 (red) was used to detect hepatic lipoperoxidation under a fluorescent microscope. Three independent experiments were analyzed. Nuclei were stained with DAPI (blue). (**B**) Fluorescent images were quantified and compared with oleic acid-induced HepG2 cells. (**C**) Oleic acid-induced HepG2 cells were treated with licochalcone A for 24 h before staining with the fluorescent probe BODIPY FL C12 (green). Nuclei were stained with DAPI (blue). (**D**) Fluorescent images were quantified and data are presented as the mean ± SEM; * *p* < 0.05, ** *p* < 0.01 compared with oleic acid-induced HepG2 cells. Three independent experiments were analyzed.

**Figure 9 cells-08-00447-f009:**
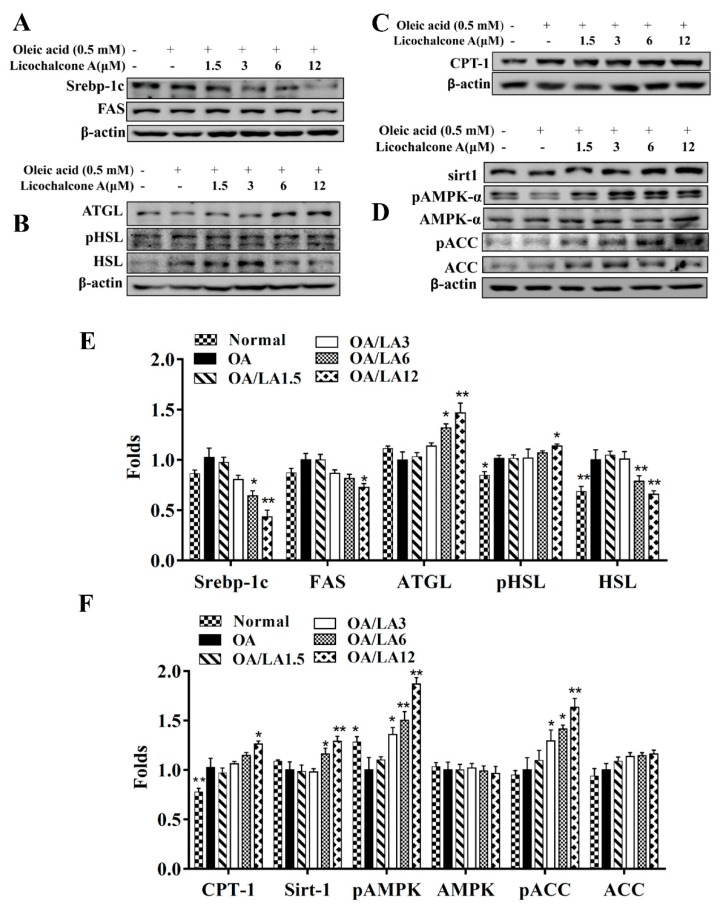
Effects of licochalcone A (LA) on lipid metabolism in HepG2cells. HepG2 cells were treated with 0.5 mM oleic acid (OA) for 48 h to induce lipid accumulation in hepatocytes, followed by treatment with licochalcone A (1.5–12 μM) for 24 h. (**A**) The expression of transcription factors associated with adipogenesis and lipogenesis proteins, (**B**) lipolysis, (**C**) β-oxidation, and (**D**) the AMPK/Sirt-1 pathway were detected by Western blot. (**E,F**) The fold expression levels were measured relative to the expression of β-actin. Three independent experiments were analyzed, and data are presented as the mean ± SEM. * *p* < 0.05, ** *p* < 0.01 compared to oleic acid-induced HepG2 cells.

**Figure 10 cells-08-00447-f010:**
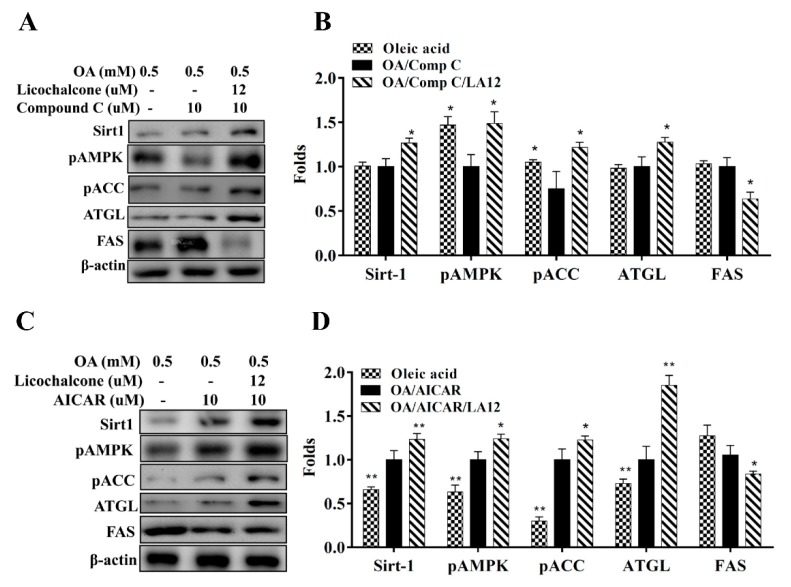
Effects of licochalcone A (LA) on AMPK/Sirt-1 pathway in HepG2 cells. HepG2 cells were treated with 0.5 mM oleic acid (OA) for 48 h, followed by licochalcone A (12 μM) with or without (**A**,**B**) an AMPK inhibitor (compound C) or (**C**,**D**) an AMPK activator (AICAR) for 24 h. Three independent experiments were analyzed using β-actin as an internal control. The images were quantified and data are presented as the mean ± SEM; * *p* < 0.05, ** *p* < 0.01 compared with oleic acid-induced HepG2 cells.

**Table 1 cells-08-00447-t001:** Primers used for real-time PCR analysis of genes.

Gene	Primer	5′–3′sequence
C/EBPα	Forward	TGGAGACGCAACAGAAGG
	Reverse	TGTCCAGTTCACGGCTCA
SREBP-1c	Forward	CTGTTGGTGCTCGTCTCCT
	Reverse	TTGCGATGCCTCCAGAAGTA
FAS	Forward	ATCCTGGCTGACGAAGACTC
	Reverse	TGCTGCTGAGGTTGGAGAG
HSL	Forward	CGGCGGCTGTCTAATGTCT
	Reverse	CGTTGGCTGGTGTCTCTGT
CPT1	Reverse	GAGACAGACACCATCCAACAC
	Forward	GAGCCAGACCTTGAAGTAACG
CPT2	Forward	TTGACCAGTGAGAACCGAGAT
	Reverse	AGAGGCAGAAGACAGCAGAG
Sirt1	Forward	CGTCTTGTCCTCTAGTTCCTGT
	Reverse	GCCTCTCCGTATCATCTTCCA
Adiponectin	Forward	GCTCTCCTGTTCCTCTTAATCC
	Reverse	ATGCCTGCCATCCAACCT
Leptin	Forward	TCAAGCAGTGCCTATCCAGAA
	Reverse	GAATGAAGTCCAAGCCAGTGA
β-actin	Forward	AAGACCTCTATGCCAACACAGT
	Reverse	AGCCAGAGCAGTAATCTCCTTC

**Table 2 cells-08-00447-t002:** Serum biochemical analysis.

	Normal	HFD	LA5	LA10
Triglycerides (mg/dL)	102.2 ± 12.7 ^*^	108.5 ± 10.4	97.2± 10.8	103.1 ± 9.8
Free fatty acid (μg/μL)	136.5 ± 25.3 ^*^^*^	181.6 ± 22.1	136.8± 17.2	113.5 ± 12.4 ^*^
Cholesterol (μg/μL)	89.7 ± 15.4 ^**^	218.8 ± 37.3	181.0 ± 10.4	168.5 ± 7.2 ^*^
LDL (mg/dL)	31.6 ± 3.3 ^**^	90.3 ± 19.1	55.0 ± 5.3 ^*^	37.1 ± 10.1 ^**^
HDL (mg/dL)	64.8 ± 8.2 ^**^	104.2 ± 14.5	131.5 ± 11.4	140.6 ± 19.5 ^*^
Bilirubin (mg/dL)	0.46 ± 0.09 ^*^	0.26± 0.02	0.36± 0.10	0.42 ± 0.09 ^*^
Leptin (μg/mL)	20.9 ± 0.7 ^**^	78.2 ± 5.7	38.9 ± 7.5 ^**^	35.9 ± 5.3 ^**^
Adiponectin (mg/mL)	2.4 ± 0.4 ^*^	2.1 ± 0.4	2.9 ± 0.3 ^*^	3.2 ± 0.3 ^**^
TNF-α (pg/mL)	19.2 ± 3.2 ^*^	49.4 ± 7.2	33.1 ± 5.1 ^*^	32.5 ± 6.9 ^*^
GPT (U/L)	80.9 ± 13.1 ^*^	140.1 ± 6.8	103.4 ± 10.9 ^*^	88.7 ± 17.1 ^*^
GOT (U/L)	48.8 ± 10.4 ^**^	102.5 ± 22.5	72.7 ± 18.1 ^*^	45.2 ± 12.1 ^**^
Glucose (mg/dL)	118.5 ± 18.2 ^**^	283.7 ± 12.6	174.2 ± 20.1 ^*^	141.3 ± 18.2 ^**^
Insulin (mg/dL)	58.0± 2.7 ^**^	204.2 ± 3.5	62.5 ± 2.4 ^**^	70.4 ± 1.9 ^**^
HOMA-IR	0.48 ± 0.09 ^**^	4.33 ± 0.58	1.54 ± 0.53 ^**^	0.84 ± 0.12 ^**^

* *p* < 0.05, ** *p* < 0.01 compared to HFD-induced obese mice. Data are presented as mean ± SEM.

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
