# Peer review of "Protective Effects of Licochalcone A Ameliorates Obesity and Non-Alcoholic Fatty Liver Disease Via Promotion of the Sirt-1/AMPK Pathway in Mice Fed a High-Fat Diet"

_cells, 2019, doi:10.3390/cells8050447_

Reviewer 1 Report

Liou et al here showed protective effect of licochalcone A from Glycyrrhiza uralensis in high fat diet induced animail model or oleic acid-treated heptocytes. Although some of the results are interesting, data is not sufficient to support your claim. Moreover, as the authors cited, Quan et al (2012) already publicated the effect of licochalcone A in high fat diet mice model and hepatocytes. The authors should elaborate to present the discrepancy with prior report. My detailed comments are the following:

1) Title, the authors claim that effect of licochalcone A is due to sirt1/AMPK activation in mice fed a high fat diet. If so, the authors should examine the effect of licochalcone A in the sirt1 KO mice.

2) Introduction is too long and redundancy. The authors should edit introduction.

3) Fig 3, The authors should add oil red o staining and hepatic TG levels.

4) Fig 4 and 9, The authors should provide graph containing a quantification of bands.

5) Fig 7-9, What is the meaning of nornal in oleic acid treated cells. Method of oleic acid preparation should be presented in the method.

6) Same as comment 1, AMPK and/or Sirt siRNA or inhibitor experiment should be added.

Author Response

Liou et al here showed protective effect of licochalcone A from Glycyrrhiza uralensis in high fat diet induced animail model or oleic acid-treated heptocytes. Although some of the results are interesting, data is not sufficient to support your claim. Moreover, as the authors cited, Quan et al (2012) already publicated the effect of licochalcone A in high fat diet mice model and hepatocytes. The authors should elaborate to present the discrepancy with prior report. My detailed comments are the following:

Responses:

Thank for reviewer’s suggestion. Quan et al found that licochalcone A could reduce the triglyceride levels in HFD-induced obese mice. However, how licochalcone A improves NAFLD and modulates the molecular mechanism of lipogenesis in obese mice is elusive. We added more discussion and explanation in the section of discussion.

1) Title, the authors claim that effect of licochalcone A is due to sirt1/AMPK activation in mice fed a high fat diet. If so, the authors should examine the effect of licochalcone A in the sirt1 KO mice.

Responses:

We found that licochalcone A also stimulated sirt1, phosphorylation of AMPKα, and phosphorylation of ACC-1 expression compared with the HFD group. But our lab doesn't have Sirt1 or AMPK knock-out mouse. So, the fatty liver cell model was established by incubating HepG2 hepatocytes with oleic acid, and treating the cells with licochalcone A to evaluate sirt1/AMPK pathway.

2) Introduction is too long and redundancy. The authors should edit introduction.

Responses:

1.Thank for reviewer’s suggestion. We modified some sentences in the introduction.

3) Fig 3, The authors should add oil red o staining and hepatic TG levels.

Responses:

We added the TC and TG levels of liver in Figure 3.  Lipid droplet morphology was observed by oil red O staining of sections from frozen tissues. But, in this manuscript, the liver tissues were fixed with formalin, and then embedded in paraffin. Those tissues did not use oil red O staining to observe lipid droplet morphology. Hence, we

used HE stain to observe fat vacuoles, and used Western blot, real-time PCR, IHC to detect lipogenesis and lipolysis in liver.

4) Fig 4 and 9, The authors should provide graph containing a quantification of bands.

Responses:

We added the quantitative result in Fig4, Fig9 and Fig10.

5) Fig 7-9, What is the meaning of nornal in oleic acid treated cells. Method of oleic acid preparation should be presented in the method.

Responses:

1. Oleic acid solution was purchased from Sigma, Inc (St. Louis, MO, USA). Hepatocyte treated with oleic acid or palmitic acid could induce lipid accumulation.

Hence, in our experiment, the fatty liver cell model was established by incubating HepG2 hepatocytes with oleic acid, and treating the cells with licochalcone A to evaluate lipid metabolism.

6) Same as comment 1, AMPK and/or Sirt siRNA or inhibitor experiment should be added.

Responses:

In Figure 10, we used Compound c (AMPK inhibitor) and AICAR (AMPK activator) to assay lipogenesis expression. We found that licochalcone A could recover the levels of phosphorylated AMPK, Sirt1, phosphorylated ACC, and ATGL and decrease FAS expression when oleic acid-induced HepG2 cells were co-treated with compound C (an AMPK inhibitor) (Fig 10A-B). Interestingly, oleic acid-induced HepG2 co-cultured with licochalcone A and AICAR (an AMPK activator) also had increased levels of phosphorylated AMPK, Sirt1, phosphorylated ACC, and ATGL and decreased FAS expression compared with oleic acid-induced HepG2 cells treated with AICAR (Fig. 10C-D)

Reviewer 2 Report

Dear authors,

In chapter Introduction provide more information about licochalcone A and it's influence on liver injury, please.

In Results it's easy to add information about influence of licochalcone A on insulin resistance - you have done glucose and insulin levels.

In Discussion please add your own opinion about the test results and the possibilities of their use in human research.

Please clearly formulate the conclusions. 

Author Response

Dear authors,

In chapter Introduction provide more information about licochalcone A and it's influence on liver injury, please.

Responses:

1.Thank for reviewer’s suggestion. We provide more information about licochalcone A and it's influence on liver injury in the introduction.

In Results it's easy to add information about influence of licochalcone A on insulin resistance - you have done glucose and insulin levels.

Responses:

We calculated HOMA-IR value in table 2. We also measured the HOMA-IR value to assess insulin resistance. Licochalcone A significantly reduced HOMA-IR value for ameliorated insulin resistance in obese mice.

In Discussion please add your own opinion about the test results and the possibilities of their use in human research.

Responses:

We confirmed that licochalcone A reduced adipose tissue and body weights, and significantly reduced lipid accumulation in the liver of obese mice by promoting the sirt1/AMPK pathway, ameliorating hepatic steatosis. Clinically, licochalcone A maybe have potential as a novel, anti-obesity agent for treating NAFLD. We added some description and explanation in Discussion.

Please clearly formulate the conclusions.

Responses:

We modified some sentences in the conclusions. Thank for reviewer’s suggestion.

Reviewer 3 Report

The study by Liou et al. investigates the role of licochalcone A (LA) in obesity and NAFLD.  The authors find that LA reduces weight gain, NAFLD and that this occurs through modulation of multiple signaling pathways including Sirt1/AMPK.  The greatest strength of this study is the in depth analysis of the transcription factors and proteins that are changed during HFD and by LA treatment.  Overall, the work is novel and important to those in the field.  That being said, there are areas the authors should address to improve the quality of their study:

1)      In the introduction, the first two sentences (page 2, lines 7-9) are unnecessary and the first could be considered offensive by some readers.  I would recommend removing these sentences.

2)      Did the authors make their own diet or was it outsourced?  The authors should include the full components of the diet.  Was it high or low carbohydrate content?

3)      Methods section 2.6, include primers used and if they were designed or commercially available.  If made, the authors need to list primer sequences in a table.

4)      Figure 1:  Did the authors measure food intake of the mice?  Is it possible that LA5 and LA10 treatment reduced appetitive behavior and thus generated their effects through less food intake?

5)      Figure 1b:  Why were the control mice already having a lower weight as a group before HFD or LA treatment?  How were the groups of mice selected?  The authors need to update the methods on how each group of mice were chosen.

6)      Table 1:  The authors should assess serum bilirubin and additional adipokines and add them to the table.

7)      Figure 4:  The authors should quantify the immunoblot images as protein fold compared to beta-actin.  This will help support the text as some of the data is difficult to the changes described in the text.

8)      Figure 7A: Left labels should be Oleic Acid + LA.  As indicated it appears that those cells were not treated with oleic acid which is not correct according to the text.

9)      Figures 7B and 8B:  What are the y-axis label referring to?  The authors need to update the methods on how they are quantifying their immunofluorescence images.  Is this % area?  If this is the case, the images provided have different numbers of cells on them and does this contribute to the results presented?  In the methods include number of cells seeded per well.

10)   Line 12 page 10:  The stain is Oil Red O.

11)   Figure 7C:  This should be quantified as it appears that the most Oil Red O staining is in the LA 3uM group.  The authors should include better labels like mentioned on comment #9.  In addition, the figure legend mentions that three independent experiments were analyzed but this appears incorrect as there was no graph present.

12)   Figure 8C:  Why was only one concentration of LA shown?  The labels need to be corrected to include Oleic acid + LA.   This figure should be quantified.

13)   Figure 9E and 9F:  Both of these figures are missing an important group.  9E should have LA without compound C.  9F should have LA without AICAR.

14)   Figure 9:  Immunoblots should be quantified.

15)   Page 13 line 45:  Methionine-choline deficient diet is generally used as a NASH model.

16)   As Sirt1 can be a positive modulator of autophagy, the authors should perform western blots or immunofluorescence for ATF5/7 and LC3 to determine if LA is generating some of its effects through autophagy induction.

17)   There are wording and grammatical errors throughout that need to be corrected.  Please proofread manuscript carefully and correct these.

Author Response

The study by Liou et al. investigates the role of licochalcone A (LA) in obesity and NAFLD.  The authors find that LA reduces weight gain, NAFLD and that this occurs through modulation of multiple signaling pathways including Sirt1/AMPK.  The greatest strength of this study is the in depth analysis of the transcription factors and proteins that are changed during HFD and by LA treatment.  Overall, the work is novel and important to those in the field.  That being said, there are areas the authors should address to improve the quality of their study:

1) In the introduction, the first two sentences (page 2, lines 7-9) are unnecessary and the first could be considered offensive by some readers.  I would recommend removing these sentences.

Responses:

1.Thank for reviewer’s suggestion. We removed and modified some sentences in the introduction.

2) Did the authors make their own diet or was it outsourced?  The authors should include the full components of the diet.  Was it high or low carbohydrate content?

Responses:

High Fat Diet (No. D12492) was purchased from Research Diets, Inc (Middlesex County, New Jersey, United States). The information added the section of Materials and Methods

3) Methods section 2.6, include primers used and if they were designed or commercially available.  If made, the authors need to list primer sequences in a table.

Responses:

We synthesized specific primers and showed in table 1

4) Figure 1:  Did the authors measure food intake of the mice?  Is it possible that LA5 and LA10 treatment reduced appetitive behavior and thus generated their effects through less food intake?

Responses:

We found that LA5 and LA10 groups did not alter food intake compared to HFD mice. The result was showed in Figure 1E.

5) Figure 1b:  Why were the control mice already having a lower weight as a group before HFD or LA treatment?  How were the groups of mice selected?  The authors need to update the methods on how each group of mice were chosen.

Responses:

Male C57BL/6 mice (4 weeks old) were procured from the National Laboratory Animal Center in Taiwan. Mice fed a high fat diet or standard diet four weeks. Then,

All mice were randomly divided into four groups of 12. Hence, fourth week of the experiment, Normal control mice (fed standard diet) having a lower weight as a group before HFD or LA treatment mice.

6) Table 1: The authors should assess serum bilirubin and additional adipokines and add them to the table.

Responses:

We added the results of bilirubin and adipokines (including leptin, adiponectin and TNF-α) in Table 2.

7) Figure 4:  The authors should quantify the immunoblot images as protein fold compared to beta-actin.  This will help support the text as some of the data is difficult to the changes described in the text.

Responses:

We added the quantitative result in Fig4 and Fig9

8) Figure 7A: Left labels should be Oleic Acid + LA.  As indicated it appears that those cells were not treated with oleic acid which is not correct according to the text.

Responses:

We modified the label as OA /LA in Figure 7.

9) Figures 7B and 8B:  What are the y-axis label referring to?  The authors need to update the methods on how they are quantifying their immunofluorescence images.  Is this % area?  If this is the case, the images provided have different numbers of cells on them and does this contribute to the results presented?  In the methods include number of cells seeded per well.

Responses:

We use the computer software of the microscope to quantify the fluorescence brightness. We modify the unit of quantitation as a percentage

10) Line 12 page 10:  The stain is Oil Red O.

Responses:

Thank you to the reviewer for pointing this mistake.

11) Figure 7C:  This should be quantified as it appears that the most Oil Red O staining is in the LA 3uM group.  The authors should include better labels like mentioned on comment #9.  In addition, the figure legend mentions that three independent experiments were analyzed but this appears incorrect as there was no graph present.

Responses:

HepG2 cells treated with isopropanol and lipid accumulation measured using the absorbance at OD 490 nm. Three independent experiments were analyzed. Three independent experiments were analyzed. The data are presented as the mean ± SEM; *P < 0.05, **P < 0.01 compared with oleic acid-induced HepG2 cells.

12) Figure 8C:  Why was only one concentration of LA shown?  The labels need to be corrected to include Oleic acid + LA.   This figure should be quantified.

Responses:

We modified the label as OA + LA in Figure 8A and Figure 8C.

Staining with the fluorescent dye BODIPY 493/503 demonstrated that incubating HepG2 cells with oleic acid induced lipid accumulation, and licochalcone A significantly inhibited the accumulation of lipid droplets in a dose-dependant manner. We also used oil red O stain to confirm that licochalcone A alleviated lipid droplets compared with oleic acid-induced HepG2 cells. Hepatic lipoperoxidation was detected by BODIPY 581/591 C11, and we found that licochalcone A suppressed lipoperoxidation compared with oleic acid-induced HepG2 cells in a dose-dependant manner. Hence, fatty acid uptake assay only used 12 uM licochalcone A treated with oleic acid-induced hepatocytes.

13) Figure 9E and 9F:  Both of these figures are missing an important group.  9E should have LA without compound C.  9F should have LA without AICAR.

Responses:

Compound c is AMPK inhibitor, and AICAR is an AMPK activator. In recent years, many studies found that Compound C is commonly used as an inhibitor of AMPK, which serves as a key energy sensor in cells. Compound C was shown to prevent the inactivation of acetyl CoA carboxylase following incubation with either AICAR or metformin. We thought that more studies had confirmed the regulation function of Compound C and AICAR in lipid metabolism. Hence, we did not show Compound C and AICAR only in Figure 10A and 10C.

14) Figure 9:  Immunoblots should be quantified.

Responses:

We added the quantitative result in Fig9 and Fig10

15) Page 13 line 45:  Methionine-choline deficient diet is generally used as a NASH model.

Responses:

Thank for reviewer’s suggestion, we modified the sentence in the menuscript.

16) As Sirt1 can be a positive modulator of autophagy, the authors should perform western blots or immunofluorescence for ATF5/7 and LC3 to determine if LA is generating some of its effects through autophagy induction.

Responses:

Lv et.al., found that licochalcone A has protective potential against LPS/ d-galactosamine -induced hepatotoxicity, which may be strongly associated with activation of Nrf2 and autophagy [1]. SIRT1 also was thought that could positively regulates autophagy and mitochondria function in embryonic stem cells under oxidative stress [2]. Sirt1 is a NAD+-dependent deacetylase that regulates intracellular NAD+ levels to maintain energy balance. Sirt1 expression also induces AMP-activated protein kinase (AMPK) phosphorylation to regulate lipid and glucose metabolism. AMPK phosphorylation could stimulate acetyl CoA carboxylase phosphorylation, reducing lipid biosynthesis. Recent research found that the Sirt1/AMPK pathway is an important regulator sensor for energy balance. Excessive lipid accumulation in adipocytes and hepatocytes will suppress the activity of Sirt1 and reduce the energy regulation capacity of AMPK. Therefore, promoting Sirt1/AMPK pathway activity would contribute to a reduction in lipid accumulation and improve hepatic steatosis. We did not investigate autophagy signal pathway in this manuscript. Thank for reviewer provide the gainful information for research NAFLD, in the future. To regulated autophagy maybe a potential as a novel anti-obesity agent for the treatment of NAFLD.

References

1. Lv. H.; Yang. H.; Wang. Z.; Feng. H.; Deng. X.; Cheng. G.; Ci. X. Nrf2 signaling and autophagy are complementary in protecting lipopolysaccharide/ d-galactosamine-induced acute liver injury by licochalcone A. Cell Death Dis. 2019, 10, 313.

2.Ou, X.; Lee, M.R.; Huang, X.; Messina-Graham, S.; Broxmeyer, H.E. SIRT1 positively regulates autophagy and mitochondria function in embryonic stem cells under oxidative stress. Stem Cells 2014 32, 1183-1194.

17) There are wording and grammatical errors throughout that need to be corrected.  Please proofread manuscript carefully and correct these.

Responses:

Thank for reviewer’s suggestion, we had checked the manuscript by San Francisco Edit as the supplement 1(Invoice No: 180033).

Round  2

Reviewer 1 Report

Although, authors could not address all my concern/comments, manuscript has been improved. This reviewer has no further concerns.

Author Response

Although, authors could not address all my concern/comments, manuscript has been improved. This reviewer has no further concerns.

Responses:

We thank the reviewers because these points were useful to clarify the main focus of the paper.

Reviewer 3 Report

The revised manuscript is much improved and the authors took time to address all my concerns.  Based upon the new data and text added, there are a few minor issues the authors should address:

1)      Lines 242-243:  Why does lipochalcone A increase HDL and bilirubin?  The authors should add additional details to the discussion regarding this finding and its implications.

2)      Are the authors measuring total or conjugated bilirubin? Please add this detail to the methods.

3)      Line 99:  ICR abbreviation needs to be defined.

4)      Line 236:  NADLF should be NAFLD

Author Response

1) Lines 242-243: Why does lipochalcone A increase HDL and bilirubin?  The authors should add additional details to the discussion regarding this finding and its implications.

Responses:

Thank for reviewer’s suggestion. We add additional discussion in the section of the discussion. (Line 491-503)

“Previous studies found that cholesterol transported into the liver would be converted to LDL and HDL[1]. Excessive accumulation of plasma LDL maybe form plaques in vascular wall for causing atherosclerosis. HDL can remove LDL and other lipoproteins from the circulation for decreasing the risk of cardiovascular disease[2]. Our result demonstrated that licochalcone A reduced TC, and LDL levels in serum of obese mice. Hence, licochalcone A can attenuate cardiovascular disease in obese mice. Bilirubin is the end product of heme metabolism. Some studies found that bilirubin could increase antioxidant capacity for improved oxidative stress-induced diseases [3]. In recent years, research identified that the bilirubin could ameliorate insulin resistance by modulating cholesterol metabolism in obese mice [4]. Our experiment demonstrated that licochalcone A had the ability to restore serum bilirubin levels in obese mice. Licochalcone A also reduced the HOMA-IR value for improved insulin resistance. We thought that licochalcone A enhanced the bilirubin levels of the serum to protect the development of insulin resistance in obese mice.”

1.Lyu , J.; Yang, E.J.; Shim, J.S. Cholesterol trafficking: an emerging therapeutic target for angiogenesis and cancer. Cells 2019, 8, E389.

2.Narwal, V.; Deswal, R.; Batra, B.; Kalra, V.; Hooda, R.; Sharma, M.; Rana, J.S. Cholesterol biosensors: A review. Steroids 2019, 143, 6-17.

3.Vítek, L. The role of bilirubin in diabetes, metabolic syndrome, and cardiovascular diseases. Front. Pharmacol. 2012 3, 55.

4.Liu, J.; Dong, H.; Zhang, Y.; Cao, M.; Song, L.; Pan, Q.; Bulmer, A.; Adams, D.B.; Dong, X.; Wang, H. Bilirubin increases insulin sensitivity by regulating cholesterol metabolism, adipokines and PPARγ levels. Sci. Rep. 2015, 5, 9886.

2) Are the authors measuring total or conjugated bilirubin? Please add this detail to the methods.

Responses:

We detect total bilirubin, and added the information in the methods.

3) Line 99: ICR abbreviation needs to be defined.

Responses:

Thank for reviewer’s suggestion. We modified the sentence as “ Licochalcone A was found that could decrease the triglyceride levels in the liver of high fat diet (HFD)-induced obese Institute of Cancer Research (ICR) mice” (Line 99-100)

4) Line 236: NADLF should be NAFLD

Responses: We changed NADLF as NAFLD (Line 237)
